# Genetic Distinctiveness Highlights the Conservation Value of a Sicilian Manna Ash Germplasm Collection Assigned to *Fraxinus angustifolia* (Oleaceae)

**DOI:** 10.3390/plants9081035

**Published:** 2020-08-14

**Authors:** Loredana Abbate, Francesco Mercati, Giuseppe Di Noto, Myriam Heuertz, Francesco Carimi, Sergio Fatta del Bosco, Rosario Schicchi

**Affiliations:** 1Institute of Biosciences and Bioresources (IBBR), National Research Council, Corso Calatafimi 414, 90129 Palermo, Italy; loredana.abbate@ibbr.cnr.it (L.A.); francesco.carimi@ibbr.cnr.it (F.C.); sergio.fatta@ibbr.cnr.it (S.F.d.B.); 2Department of Agricultural, Food and Forestry Sciences (SAAF), University of Palermo, Via Archirafi 38, 90123 Palermo, Italy; pippo.dinoto@virgilio.it (G.D.N.); rosario.schicchi@unipa.it (R.S.); 3Institut National de Recherche Pour l’agriculture, l’alimentation et l’environnement (INRAE), Univ. Bordeaux, BIOGECO, 69 route d’Arcachon, F-33610 Cestas, France; myriam.heuertz@inra.fr

**Keywords:** *Fraxinus* spp., manna, local varieties nSSR, cpSSR, cytometry, morphological traits

## Abstract

The cosmopolitan genus *Fraxinus* comprises about 40 species occupying several habitats in the Northern Hemisphere. With some species hybridizing and sharing genetic variants, questions remain on the species assignment of germplasm within the genus *Fraxinus* despite numerous species-specific assessments. A multidisciplinary approach was employed to provide a definitive insight into the genetics of an endangered *Fraxinus* “manna ash” collection, located in a rich plant biodiversity hotspot of the Madonie Mountains (Sicily). Although the collection size was small, genetic diversity, assessed by chloroplast (cpSSR) and nuclear (nSSR) microsatellites (SSR—Simple Sequence Repeats), allowed identifying three different chloroplast haplotypes, with one (H5) dominant, and several polymorphic loci, able to discriminate most of the local accessions studied. Molecular data were linked to cytofluorimetric and phenotypic evaluations and, contrary to popular belief that manna ash is *Fraxinus ornus* L., the germplasm currently used for manna production belongs to *Fraxinus angustifolia* Vahl. Interestingly, joint analysis of our genetic panel with a large European dataset of *Fraxinus* spp. suggested the presence of a possible glacial refuge in Sicily, confirming its importance as biodiversity source. Our results will be helpful for the design of long-term conservation programs for genetic resources, such as in situ and ex situ conservation, seed collection and tree reintroduction.

## 1. Introduction

The genus *Fraxinus* (Oleaceae) comprises 45–65 tree species and is represented across large areas of Europe, Asia and North America [1,2]. *Fraxinus* species show considerable variability in their flowering biology, ecological requirements and distribution ranges as a result of dispersal and vicariance processes as well as adaptive evolution underlying diversification in the genus [3].

In Europe, three ash species are present: the common ash, *Fraxinus excelsior* L., the flowering ash, *Fraxinus ornus* L., and the narrow-leaved ash, *Fraxinus angustifolia* Vahl [4]. *F. excelsior*, a polygamous species with male, female and hermaphrodite individuals, [5,6], is found throughout the continent except in the Mediterranean region, while *F. ornus* is androdiecious [3] and grows in relatively high altitude areas of the eastern Mediterranean basin [7,8]. Finally, *F. angustifolia*, a close relative to *F. excelsior*, shows hermaphrodite flowers and is very common around the Mediterranean basin and to the west of the Black Sea in the Danube basin. Due to the level of local adaptation [9], different botanical names were assigned to *F. angustifolia.* Three groups can be identified at subspecies level, structured by geographical regions [7,10]: *F. angustifolia* ssp. *oxycarpa* (Bieb. ex Willd.) Franco and Rocha Afonso observed in the in East Central and Southeastern Europe, including the Balkans; *F. angustifolia* ssp. *syriaca* (Boiss.) Yalt. in the east of Europe, from Turkey to Pakistan; and *F. angustifolia* ssp. *angustifolia* in southwestern Europe. While *F. excelsior* grows on a wide range of soil types, preferring nutrient-rich substrates, *F. angustifolia* grows near surface and ground waters, and thus its habitat is more restricted [11,12]. Indeed, the distribution of this species in the Mediterranean region is irregular and limited to smaller and isolated populations on drier sites at higher altitudes or on wetland sites [13]. The levels and distribution of genetic diversity in *F. excelsior* and *F. angustifolia* reflect the differential climatic and ecological affinities of both species, as well as the signature of ancient and contemporary hybridization [4,14].

Sicily boasts several botanical highlights. Both trees and endemic herbs survived through the last glacial maximum, highlighting the role of Sicily as a refuge area for thermophilous European plants during glacial times [15]. In addition Sicily, the biggest region of Italy, has been identified as one of 52 putative glacial refuges based on phylogeographic data from trees and herbs, and lies within one of ten regional hotspots of plant species biodiversity in the Mediterranean basin [16].

The presence of the genus *Fraxinus* in Sicily is restricted mainly to scattered groves and natural populations of *F. angustifolia* and *F. ornus* spread in the hilly area of the Madonie and Nebrodi Regional Natural Parks, at altitudes between 100 and 900 m a.s.l. In this area, the relevance of *Fraxinus* spp. is extremely high, from economical, botanical, naturalistic and historical points of view [17,18]. In fact, the *F. ornus* cultivation, brought to Sicily at the time of the Arab invasion (ninth-eleventh century) and known as manna ash, is grown for the production of manna, a crystal coagulate exuding after cutting 5–10 cm long slits with a billhook along and inside the whole thickness of the bark. From these slits a purplish and bitter liquid flows out. After contact with the air, this liquid turns white and becomes sweet, rapidly coagulating and forming crystal-like layers, called manna. Then manna is harvested and placed in specific trays for drying. Its production varies from 0.2 to 1.5 kg per tree per year, depending on ash cultivar [18]. However specific pedoclimatic conditions (persistent dry, hot periods and steady ventilation during summertime) are mandatory for the bioaccumulation of osmotic sugars in the phloem of *Fraxinus* and to obtain the complete drying of the sap that exudes from cuttings. The most abundant constituent of manna is mannitol (around 50% of total chemical compounds), a hexavalent alcohol known also as “manna sugar”; monosaccharide, oligosaccharide, mineral and volatile constituents are also well represented [18,19]. Due to its characteristics, the manna is used in cosmetic, pharmaceutical and confectionary industries [20]. In Sicily, the presence of *Fraxinus* groves has given rise to a traditional agroforestry landscape of unique value and relevant economic importance until the late 1950s. In the last 50–60 years, after the introduction of synthetic mannitol, the integrity and the extension of the manna ash landscapes have been dramatically reduced. Today, because of habitat fragmentation, naturally poor regeneration and land use change, the maintenance of the crop is threatened. The present landscape of Sicilian manna *Fraxinus* is limited to a fragmented and isolated area of few hundreds of hectares in the territory around Castelbuono and Pollina (Palermo-Italy), in which natural populations, single and scattered trees are distributed. Large differences in several phenotypic traits, such as leaf morphology, inflorescence features and infructescence characteristics, are observed among the different plants. However, based on their distinctive morphological features, several cultivars (or varieties) were distinguished and identified with dialect names, since the nineteenth century [17,20,21,22]. The declining status of the manna ash germplasm has been confirmed by the last survey on the residual consistency of the crop in the Madonie Mountains [17,18]. In less than 150 years, about 50% of the germplasm of the manna ash cultivars has been lost; therefore, both the germplasm of the cultivated varieties and the ash cultivation are threatened with extinction. Previous studies, based on morphological and chemical analysis, confirmed the presence of 16 cultivars, 13 of which putatively belonging to *F. angustifolia* and the remaining three putatively belonging to *F. ornus* [17]. However, the value of these approaches to characterize manna cultivars and varieties is limited due to their drawbacks, such as the influence of environment on trait expression, epistatic interactions and pleiotropic effects. Therefore, there is an urgent need to focus on the unknown genetic background, structure, relationships and diversity of the *Fraxinus* gene pool distributed in Sicily, to provide useful information aimed both to prevent biodiversity loss, safeguarding this endangered plant populations, and develop future breeding programs.

Microsatellite (SSR, simple sequence repeat) markers are codominant, highly polymorphic, reproducible, well distributed throughout the genome and suitable for automated analysis, and are; therefore, widely used markers for plant variety characterization [23,24,25,26,27,28]. SSRs have proved to be ideal tools, in conjunction with analysis of morphological characteristics, for analyzing genetic diversity and structure and evolutionary relationships between populations of rare, isolated and endangered species. Chloroplast DNA sequences are suitable tools to highlight the postglacial colonization routes [4,29,30] because they are usually non-recombinant in angiosperms and they are transmitted through seeds only [31,32], implying that the effective population size is more reduced for chloroplast DNA than for nuclear DNA. Nuclear and plastid DNA microsatellites have been extensively used to describe the phylogeography of *Fraxinus* spp. and clarify the hybridization process among ash species [4,14,33,34].

Nuclear DNA content has an important role in systematics, and it is a useful tool in biodiversity evaluation [35,36]. Flow cytometry is a quick and effective method to assess genome size (i.e., the amount of DNA in cell nuclei) [37,38]. Studies of genome size variation can shed light on the molecular mechanisms underlying this type of variation [39,40,41], which is a relevant indicator of genetic divergence and speciation processes [42,43].

In this study, we used morphological traits, genetic profiles (nuclear and plastid SSR markers) and genome size estimates (nuclear DNA content) to characterize the biodiversity within and among native accessions of *Fraxinus* in Sicily, belonging to a historical manna ash collection that, from previous studies [18], seems to be represented by varieties mainly characterized as *F. angustifolia*. We also present the first assessment of evolutionary relationships between the *F. angustifolia* germplasm from Sicily and *F. angustifolia* germplasm from throughout Europe. Knowledge of the extent of genetic diversity and characterization of population genetic structure of native *Fraxinus* spp. in Sicily will have important implications for the conservation of this biodiversity, to provide useful knowledge for the management of the manna ash germplasm collection, in-situ/ex-situ conservation and to save this precious cultural heritage.

This analysis is a first step towards selecting breeding material and establishing conservation strategies to sustain and potentially increase the overall production and productivity of manna ash and the continual use of this agroforestry tree resource through varietal improvement and suitable agronomic practices under southern Mediterranean conditions.

## 2. Materials and Methods

### 2.1. Plant Material

Leaf samples of native manna ash (*Fraxinus* spp.) germplasm located in Madonie National Park (Palermo, Sicily; 37°53′ N 14°01′ E/37.883° N 14.017° E) were collected. Individual trees were selected according to specific criteria based on morphological features, historical notes and their use for manna production [17,18,21,22,44]. The list of accessions studied, and their dialectical name, is indicated in Table 1 and Appendix A. At least three individual trees (replicates) were collected for each local variety studied, except for monumental trees. In total thirty-four (34) trees (accessions) were analyzed. Four *F. ornus* trees, collected in the same area, were also added in the study and used as reference. Young and fully expanded leaves were immediately frozen in liquid nitrogen and conserved at −80 °C until used for DNA extraction. This sample set, comprising 38 accessions (Appendix A) belonging to manna ash varieties and *F. ornus* (FOR) collected in Sicily, is named sample set #1 from here on.

To full investigate genetic relationships and population genetic structure among the analyzed manna ash accessions and natural populations of *Fraxinus* spp., our sample set (#1) was added to SSR profiles already available. To avoid redundancy in the results, this second sample set (#2) included only samples showing unique genetic profiles isolated from sample set #1, and SSR unique profiles of 475 individual ash trees described in Gérard et al. (2013) [14], belonging to *F. angustifolia* (148; FAN), *F. excelsior* (274; FEX) and hybrids between *F. angustifolia* and *F. excelsior* (53; HYB) collected in the natural, partially overlapping, distribution ranges of the two species across Europe (Appendix A). This second sample set is named sample set #2 from here on.

### 2.2. Morphological Trait Variation

For each accession from set #1, including four *F. ornus* trees, a set of morphological traits were evaluated (Appendix A). The most discriminant morphological features among *Fraxinus* species (e.g., the samara stalk length, the number of samaras/raceme, the colour of apical buds) were recorded; three replicates for each trait were evaluated.

### 2.3. Genome Size and Ploidy Level

The genome sizes of collected accessions (Table 1) were estimated by flow cytometry using the *F. ornus* (2C-value = 1.98 pg) as reference standard. Nuclei isolated from a single mature leaf were analyzed in three technical replicates for each accession of set #1. The analysis was carried out with a Partec PAS flow cytometer (Partec, http://www.partec.de/), equipped with a mercury lamp. Fully expanded leaves (1 cm^2^) were chopped in a glass Petri dish with 1 mL nuclei extraction buffer OTTO1 [45] and 3 drops of Tween 20. After 3 min, 1 mL of OTTO2 [45] supplemented with DAPI (4 µg/mL) was added. The solution was filtered through a 30 μm Cell-Trics disposable filter (Partec). The relative fluorescence intensity of stained nuclei was measured on a linear scale, and 4000–5000 nuclei for each sample were analyzed. DNA content histograms were generated using the Partec software package (Partec-FlowMax^®^, Münster, Germany). 

### 2.4. DNA Extraction and Genotyping

Genomic DNA was extracted from 100 mg of powdered, frozen, young leaf tissue of each individual from sample set #1 using the QiagenDNeasy Plant Mini Kit (Qiagen, Hilden, Germany). The purity and quantity of the DNA extracts were assessed with a NanoDrop 1000 spectrophotometer (Thermo Scientific, Waltham, MA, USA). Molecular investigations were carried out by amplifying 6 plastid (chloroplast) microsatellite (SSR, simple sequence repeat) markers (cpSSR) (ccmp2, ccmp3, ccpm4, ccmp6, ccmp7, and ccmp10 [46]), 3 plastid DNA regions (atpB-rbcL, CPFRAX6 and matK [33]) and 11 nuclear microsatellite nSSR (FEMSATL4, FEMSATL11, FEMSATL12, FEMSATL16, FEMSATL19 [47], M230 [48], EST-SSR326, EST-SSR427, EST-SSR431, EST-SSR520, EST-SSR528 [49]) markers, respectively. PCR reactions were performed following the procedures reported in Garfì et al. (2013) [50], using primers fluorescently labelled with FAM, VIC, NED and PET. The fragments were separated by capillary electrophoresis using an ABI PRISM 3130 Genetic Analyzer (Applied Biosystems) to detect length polymorphism only. Fragments were sized and binned into alleles using Gene Mapper v. 4.1 software (Thermo Fisher Scientific, Waltham, MA, USA), (Applied Biosystems) (Appendix A).

### 2.5. Data Analysis

Plastid DNA haplotypes were defined based on the concatenation of fragment length polymorphism at cpSSRs and amplified plastid DNA regions. To assign the species of origin of the sampled materials, the obtained haplotypes were compared visually with previously published cpSSR profiles from the three European ash species reported in Heuertz et al. (2006) [4].

Genetic relationships between nSSR profiles of both sample sets (set #1; set #2) were estimated using Bruvo’s distance [51] in the poppr package [52] in R Core Team (2020; http://www.R-project.org). A dendrogram was computed from each distance matrix using the UPGMA (Unweighted Pair Group Method with Arithmetic Mean) implemented in the adegenet package [53] in R. Bootstrap analysis was performed based on 1000 replicate samples to assess the robustness of the inferred evolutionary relationships in the dendrogram.

After establishing that Sicilian manna ash is *F. angustifolia* based on the listed analysis, we assessed the genetic differentiation between Sicilian manna ash and the collection of European *F. angustifolia*, *F. excelsior* and their hybrids using Wright’s fixation index (*Fst*) [54] and Nei’s (1973) distance [55] computed through hierfstat [56] in R. The genetic variability across *F. angustifolia* (FAN accessions belonging to set #2 was estimated using the observed (*Ho*) and expected (*He*) heterozygosities [55], Shannon’s index (*I*) and the inbreeding coefficient (*Fis*) for sampling locations in GenAlEx 6.5 [57]. Each locus and population was tested for Hardy-Weinberg equilibrium deviation with the exact test through Genepop v. 3.4 [58], with the default parameters (dememorization = 10,000, number of batches = 100, and number of iterations/batch = 5000). To compare the genetic diversity of population studied with different sample sizes the allelic richness (*AR*) within sampling locations was also evaluated though SPAGeDi software [59]. Population pairwise fixation index (*Fst*) values were then computed for only *F. angustifolia* accessions, grouped based on their sampling location or country and a neighbor-joining (NJ) tree using was developed using the adegenet package [53].

Finally, to evaluate the genetic structure of *F. angustifolia* distributed across Europe (belonging to set #2 a discriminant analysis of principal components (DAPCs) was employed. Samples belonging to *F. angustifolia* were grouped based on their geographic area of origin (in five main groups: Balkans, Eastern Europe, France, Italy and Portugal) and DAPC analysis, implemented in the adegenet package [53], was carried out to infer population subdivision of the germplasm studied. The number of principal components (PCs) retained was evaluated using the cross-validation procedure. The *K*-means algorithm “find.clusters” was used to independently verify the assignment of individuals to clusters.

## 3. Results

Ploidy level, DNA content and morphological trait variation of the manna ash collection DNA content of the 34 accessions belonging to the manna ash collection and of four *F. ornus* individual trees (set #1) was estimated based on the relative intensity of fluorescence using the known *F. ornus* genome size (1.98) as reference [60]. DNA content estimates were congruent with all individuals of set #1 being diploids (2*n* = 2 × = 46) [61]. Flow cytometry-derived genome sizes (pg/2C) for manna ash varieties ranged from 1.539 to 1.544 (Table 1).

Broad variability in several phenotypic traits related to leaf morphology, inflorescence and infructescence features were observed among the different local varieties belonging to the Sicilian manna ash germplasm collection (Appendix A). Leaf shape, number of leaflets, leaflet margin, petiole length and rachis wings were highly variable (Appendix A). Also the inflorescences showed a great range of variation. Differences between varieties were mainly observed in the structure of reproductive organs, including the terminal clustering of multiple carpels, the partial basal fusion of individual carpels and the attachment and orientation of the seeds (Appendix A). The mean values of some morphological traits showed most high variability between the manna ash collection and the *F. ornus* reference trees (Table 1). Specifically, the largest phenotypic variability was found in the number of samaras/raceme (5.8 vs. 46.0), followed by samara stalk length (1.1 vs. 0.4) and apical leaf length/width ratio (0.28 vs. 0.52) (Table 1). Moreover, flowering time (November–January for manna ash vs. May–June for *F. ornus*) and the colour of apical buds (red in manna ash vs. white in *F. ornus*) were clearly distinct for manna ash and *F. ornus* (Table 1).

### 3.1. Genetic Variation of Plastid DNA Microsatellites (cpSSRs) and Amplified Regions in the Manna Ash Collection

Five (ccmp3, ccmp4, ccmp7, atpB/rbcL and matK) out of nine plastid DNA markers were monomorphic in set #1, displaying fragment lengths of 97, 140, 117, 157 and 253 bp, respectively. The other loci showed low but significant levels of polymorphism (Table 2): Three size variants, with amplification fragment sizes of 364, 365 and 366 bp, were observed at CPFRAX6, whereas two distinct size variants separated by one and four nucleotides were showed at ccmp6 and ccmp2, respectively. Finally, ccmp10 displayed three specific size variants, with amplicons of 103, 104 and 106 bp (Table 2), respectively. The size variants combined into a total of 4 haplotypes (Table 2). Except for “Verdello”, the samples belonging to each local variety showed one specific haplotype. The four haplotypes detected in set #1 represented three of the twenty-two previously characterized cpSSR-based haplotypes [4]: Manna ash accessions carried either a sub-variant of H5 (92% of accessions) or haplotype H10 (8% of accessions), these haplotypes having previously been observed in *F. angustifolia* and in *F. excelsior*; the *F. ornus* reference trees carried H19, a haplotype private to *F. ornus* (Table 2).

### 3.2. Genetic Diversity of Manna Ash Germplasm

The genetic diversity of the manna ash germplasm and *F. ornus* genotypes, the so called set #1, was investigated using 11 nSSR (Appendix A). Phylogenetic analysis based on Bruvo’s distance and the UPGMA algorithm generated a dendrogram that comprised four main clusters across set #1 (Figure 1). A total of 20 unique SSR profiles were detected (16 from manna ash collection and 4 for *F. ornus* accessions, respectively), with “Frassino monumentale” as the most distant sample among the manna ash genotypes. Except for varieties “Baciciu”, “Macigna” and “Nsiriddu”, all accessions were assigned to their expected local variety (Figure 1), showing “Verdello” as the local variety with the highest genetic variability. In addition, the accessions analyzed belonging to “Sarvaggio”, “Nivuru”, “Russo” and “Abbassa cappeddu” can be considered clones, respectively, showing the same genetic profile within the same variety. The four reference trees of *F. ornus* grouped together in a cluster that behaved like an outgroup.

### 3.3. Genetic Diversity of Fraxinus spp. in Europe

In order to compare the genetic relationships and diversity between the manna ash collection (sample set #1) and the germplasm of *F. angustifolia* and *F. excelsior* throughout the partially overlapping natural distribution ranges of both species across Europe, we combined our SSR profiles of manna ash with the genotypic profiles described in Gérard et al. (2013) [14] into a second dataset called sample set #2, which included 495 unique profiles at 11 SSRs. UPGMA analysis based on Bruvo’s distance [51] identified groups based on species (Figure 2; Appendix A). Indeed, three main clusters were highlighted, with two branches represented almost exclusively by *F. angustifolia* and *F. excelsior* samples, respectively. As expected, samples belonging to *F. ornus* clustered as the most distant genotypes, behaving as outgroup, while the individuals morphologically classified as hybrids were distributed across the two main groups. Interestingly, *F. angustifolia* samples coming from Southern Italian regions from Gérard et al. (2013) [14] data clustered together with genotypes of our manna ash collection (Figure 2), highlighting a distinctive, basal position of Southern Italian *F. angustifolia* germplasm in comparison with the range-wide collection of the species.

Interestingly, within FAN, samples collected in Southern Italy (Sicily and Calabria) showed lower genetic distance (Nei, 1973 [55]) and lower *Fst* values with respect to the other FAN populations collected in Europe, whereas comparison to FEX populations resulted in higher values (Table 3). Hybrids of both ash species were closer both to FAN (*Fst* = 0.030; *Nei* = 0.159) and FEX (*Fst* = 0.028; *Nei* = 0.110) than to FAN (Italy) (*Fst* = 0.107; *Nei* = 0.536) (Table 3).

Different genetic parameters (I, He, Ho, Fis, AR and Fst) were also estimated for *F. angustifolia* sampling locations belonging to set #2 (Table 4). Hungary and France2 locations had, respectively, the lowest (0.504) and highest (0.680) values for genetic diversity (He). Overall, genetic diversity parameters (I, He, Ho) displayed similar values across the locations studied (Table 4). Sixteen out of twenty-one locations showed an excess of heterozygotes, showing a negative inbreeding coefficient (*Fis*). However, except Hungary population (−0.607), *Fis* was close to zero therefore all groups could be considered in equilibrium (Table 4).

Overall allelic richness (*AR*) ranged from 1.173 (SSR431) to 3.746 (Fem12) for SSR431 and Fem12 markers, respectively (Appendix A). *AR* was higher in trees collected in Turkey (3.55) than the other populations. Macedonia, Montenegro, Portugal1, five out of seven French groups and the two population collected in Serbia showed similar *AR* values (ranging from 3.05 to 3.42), while population belonging to Hungary had the lowest value (2.28; Table 4).

The *Fst* values were also used to assess the genetic relationships across the population belonging to the *F. angustifolia* collection here studied (Figure 3; Appendix A). Three main clusters were observed in the NJ-tree: six out of seven populations collected in France were gathered together with the two Italian groups and Portugal2 population; except the samples collected in Croatia, the other Balkan populations clustered with samples belonging to Turkey and Hungary; and, finally, the remaining populations of France (France7), Balkans (Croatia1 and 2) and Portugal (Portugal1) were linked to Bulgaria and Ukraine populations (Figure 3). On the other hand, comparing the *Fst* recorded on samples grouped by country, the greatest distance was observed between the pair Hungary–Bulgaria; except Croatia, the populations belonging to Balkan clustered close in the NJ-tree. Portugal and France were genetically separated, with the last population near the branch that gathered Turkey and Italy (Appendix A).

DAPC identified five clusters corresponding to the five groups selected (Figure 4). Clusters including the samples belonging to France and Portugal were overlapping, as well as clusters containing Balkans and Eastern European genotypes. On the contrary all samples collected in Italy (from Calabria and Sicily) were separated from the other groups. Except for some samples, the plot of the first two principal components distinguished clearly three groups represented by genotypes from Eastern, Western and Mediterranean Europe, respectively (Figure 4).

## 4. Discussion

The genetics of endangered *Fraxinus* spp. populations is of great interest for both conservation and evolutionary aspects. Using a multidisciplinary approach, including molecular analysis, we demonstrated in this study that an important and residual manna ash germplasm collection located in the Madonie Mountains (Sicily, Italy), belongs to *F. angustifolia*, the narrow-leaved ash. We also showed that this population contained high genetic diversity and, together with a population from Calabria, was strongly differentiated from *F. angustifolia* populations from other locations in Europe (Balkans, Eastern Europe, France, and Portugal). The assessment of the genetic diversity is crucial to plan activities aimed to conservation of this important and endangered forest tree genetic resource.

Notwithstanding the limited number of individuals studied, plants belonging to the manna ash collection analyzed showed a wide variation in several phenotypic traits, such as leaf characteristics, inflorescence and infructescence features, plant vigor and morphology, showing a continuous and narrow range of values for individual traits. Leaf morphology is the strongest differentiating character within the local collection, although this trait is not successfully used to characterize the varieties, being strongly influenced by environmental factors. Indeed, it is well known that plants react to water stress, which is particularly strong in the Mediterranean area, by modifying their leaf traits. Considering that the trees collected come from a heterogeneous mountain environment in which neighboring individuals compete for the same ground and light resources, the observed phenotypic variations might be attributed to the impact of environmental factors on growth habit. Regarding fruit morphology, the shape and the length of the fruits of “manna” varieties were quite heterogeneous, as well as the colour or samaras changing from light green of most cultivars to veined brown of “Nsiriddu”. These traits; thus, showed a weak distinguishing power [62]. Nevertheless distinctive morphological traits separated all manna ash varieties from *F. ornus*. Specifically, in the local varieties the samara stalk length was greater (nearly three times) than in *F. ornus*, while the apical leaf length/width ratio and the number of samaras/raceme of manna ash samples were approximately two and a half and eight times lower, respectively, separating the two groups compared. Variety identification based on morphological data can be verified using molecular analysis, because the environmental effects, epistatic interactions and pleiotropic effects can interfere with morphological traits evaluation, as in many forest trees [63].

The genome size showed a small change in the 2C nuclear DNA content within the samples belonging to the manna ash collection (from 1.539 to 1.544 pg). The existence of intraspecific variation in genome size has been reported in several plant species [64,65,66]. However, due to the little shift here reported, the genome size, as well as the ploidy level, can be considered stable in our collection, as commonly observed within the species [67]. The variations recorded can be explained by technical issues, as reported for some angiosperms [68,69]. The 2C DNA values observed in the manna ash collection unambiguously assign the manna germplasm to *F. angustifolia* (1.540 pg [60]) and are different from both *F. excelsior* (1.68 pg; [60]), and *F. ornus* (1.98 pg; [60]), showing a larger genome size due to probably a high number of repetitive elements and/or several ribosomal gene repetitions [60].

Molecular analyses confirmed the differences highlighted through morphological and cytological approaches. In agreement with the low chloroplast DNA mutation rate detected in the Oleaceae [70], cpSSRs and plastid DNA regions identified a low genetic diversity in the manna ash collection, identifying three haplotypes, two sub-variants of H5 (covering 92% of samples) and H10 (the remaining 8%), respectively. The haplotype detected in the reference trees (H19) was previously found only in *F. ornus*, specifically in trees from Italy and Corsica. In agreement with the increased botanical similarity of *F. angustifolia* and *F. excelsior* and a partially overlapping phenology [6,71,72], an extensive sharing of cpDNA haplotypes between *F. angustifolia* and *F. excelsior* has been found previously but without common profiles between them and *F. ornus* [4]. Indeed, both H5 and H10 variants have been shown to be shared haplotypes between *F. angustifolia* and *F. excelsior* although in different geographic regions [4,73]. The majority of manna ash accessions carried haplotype H05, a haplotype observed in *F. angustifolia* throughout central and southern Italy and in *F. excelsior* in a restricted area of the Eastern Alps; this particular distribution could be related to two possible different and independent glacial refuges of ash species [73]. The other haplotype (H10) observed in “Abbassa cappeddu” variety and in some trees belonging to “Verdello” was detected in *F. angustifolia* trees collected in Portugal and Corsica [4] and only in the Czech Republic for *F. excelsior*.

Microsatellites are reported as very effective marker in terms of high information content and discrimination power owing to high allelic variation and allowing clear identification of populations or varieties in several plant species [74,75,76,77]. Nuclear SSR (nSSR) profiles showed a noteworthy rate of polymorphism and, except for “Baciciu” and “Cavolo”, were able to group each accession with its own variety. The most represented local variety named “Verdello” was grouped in a main cluster, together with tree belonging to “Nsiriddu”, showing a high variability, in agreement to the profiles obtained in natural populations of *F. angustifolia* in Greece [78]. Among studied varieties, “Verdello” and “Nsiriddu” showed an upright growth habit and are considered the most productive ones [18], yielding high quality manna with very similar chemical and organoleptic characteristics.

Although only a slight genetic differentiation between *F. angustifolia* and *F. excelsior* was reported [79,80], our findings highlighted a clear separation between trees belonging to two species collected throughout their natural distribution across Europe. Significant patterns of distance were found among species, with *F. ornus* clearly behaving like outgroup. Interestingly, a private branch grouping only *F. angustifolia* trees collected in the South of Italy, including the profiles recorded for manna ash collection, was highlighted. In agreement to cluster analysis, the pairwise fixation index (*Fst*) highlighted a clear distinctness for the populations from Southern Italy, close to Turkish and French populations. In addition both cluster and *Fst* analysis allowed to separate the two Italian populations highlight the feature belonging to manna ash collection. Our results are consistent with *Fraxinus* spp. distribution, indeed common ash (*F. excelsior*) is found mainly in the north regions throughout Europe [4,81] and has expanded its range in the south probably during cooler climate episodes, maintaining the current relic populations in temperate locations such as the Elburz Mountains (Turkey), Calabria (Italy), and in Sicily, more specifically in the Nebrodi Mountains, where a small population (200 plants) of *F. excelsior* spp. *siciliensis* [82,83] is located in three different sites (Caronia, Longi and Alcara Li Fusi), showing specific features (e.g., reduced size and blooming at the same time of leaf emission; [84] due to the high isolation. On the contrary, *F. angustifolia* is mostly restricted to the Mediterranean region [81] and the favorable environmental conditions allowed this species to spread in the Sicilian Nebrodi Mountains, giving rise at the end of 19th century more than 16 varieties distinguished for both morphological traits and manna production [21,85]. The remarkable diversity and strong genetic structure in the Sicilian collection of *F. angustifolia* highlighted by both cp and nSSRs probably is related to the geographic isolation in which they have been found for about 10,000 years starting from the end of the last glaciation. This trend of speciation in *Fraxinus*, related to specific geographic zones, is in agreement to previous work [71], and would be driven by geological and climatic modifications [86] as reported for other species [87]. Phylogenetic analysis showed also a distribution of hybrids (HYB) within both *F. excelsior* and *F. angustifolia* groups, reflecting the sharing of haplotypes between the two species [4]. More interestingly, in the last group the hybrids and trees belonging to common ash were linked to *F. angustifolia* trees collected in Balkan and Eastern Europe, in agreement to the species distribution [4], whereas the samples from Southern Italy were the most genetically distant plants. Our finding can be explained by the marked different flowering time that characterizes the two species in Sicily making the interspecific cross unfavorable.

Finally, grouping the *F. angustifolia* trees studied on the grounds of their geographic area, discriminant analysis based on nuclear microsatellites underlined a gradient from west to east in *Y*-axis. This evidence is consistent with the classification of *F. angustifolia* that can be grouped into three geographic subspecies: (i) *F. angustifolia* Vahl ssp. *angustifolia* (Portugal and the Western Mediterranean), (ii) ssp. *oxycarpa* (M. Bieb. ex Willd.) Franco and Rocha Afonso (Northeast Spain to Turkey), and (iii) ssp. *syriaca* (Boiss.) Yalt (Turkey and Asia Minor) [6,7,8]. Additionally, samples belonging to Calabria and Sicily were separated from the others. The clear diversity harbored by *F. angustifolia* populations from South Italy, allow to hypothesize a possible glacial refuge of species in this area, according to a temperature increasing during the late glacial maximum (18,000 years bp) in the Mediterranean eastern compared to western one, as already reported for the Turkish populations [4,88].

## 5. Conclusions

In summary, using a multidisciplinary approach the present study was focused to characterize in depth a historical manna ash collection represented by local varieties collected in the Madonie area (Sicily, Italy), in order to safeguard an important cultural heritage. Our finding definitively clarifies that the local varieties actually used for manna production belong to *F. angustifolia*. This study provides useful information for germplasm management, finalized to improve the production and productivity of agroforestry species investigated. In addition, the evidences here reported suggest the presence of an additional glacial refuge for *F. angustifolia* in Italy, confirming the importance of Sicily as source of biodiversity. Furthermore, our study could represent an invitation for botanists to expand the historical knowledge of the collection. Due to the importance of ash germplasm studied, all local varieties were propagated in two repository fields, at the Professional Institute for Agriculture and the Environment (I.P.A.A.) “Luigi Failla Tedaldi” (Castelbuono, Italy) and at the private Schicchi’s repository located in the Croce-Foresta district (Castelbuono, Italy) (info at rosario.schicchi@unipa.it) making available this genetic resources for future breeding programs and to develop future national/international collaborations.

## Figures and Tables

**Figure 1 plants-09-01035-f001:**
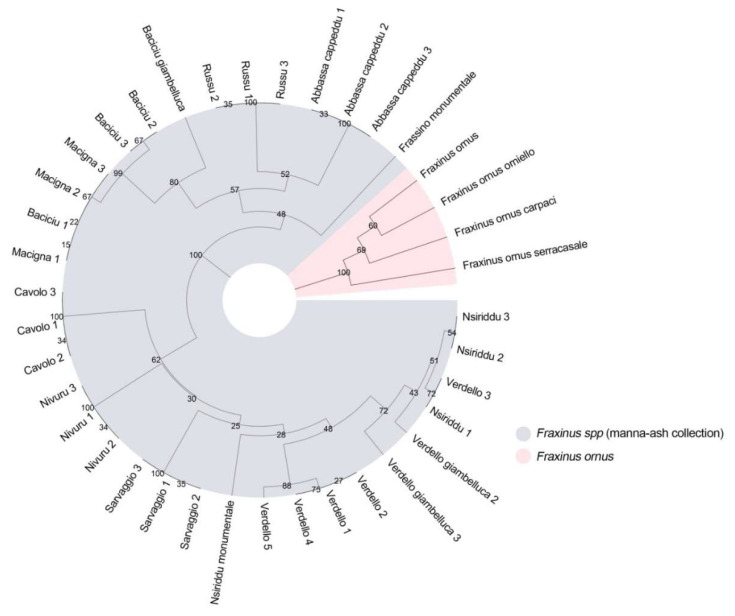
Genetic relationship developed using the profiles obtained from eleven nSSR (Appendix A) among studied local varieties (34 samples) collected in Sicily, belonging to the manna ash germplasm. Four genotypes belonging to *F. ornus* were used as references. The dendrogram was developed using the UPGMA and Bruvo’s distance [51].

**Figure 2 plants-09-01035-f002:**
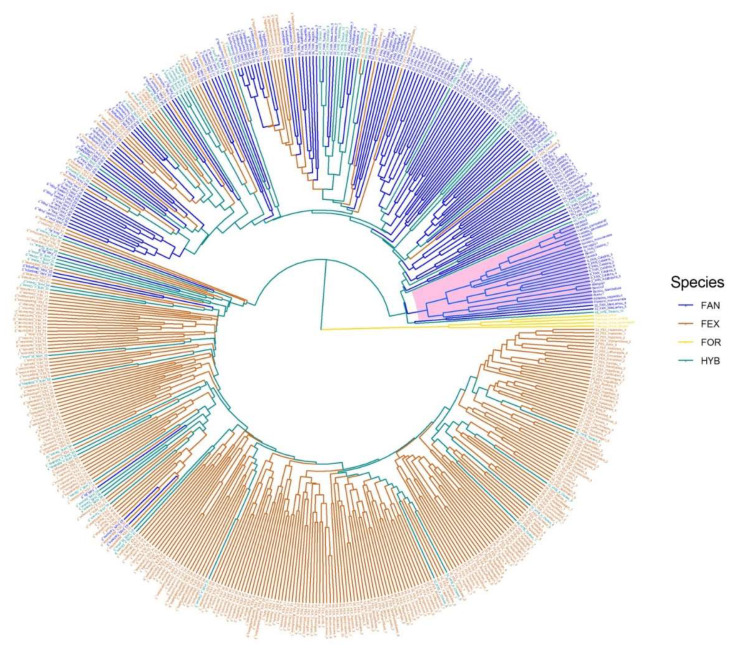
Genetic relationships among *Fraxinus* spp. accessions-based UPGMA and Bruvo’s distance, developed using unique nSSR profiles of sample belonging to set #2, including *F. angustifolia* (FAN), *Fraxinus excelsior* (FEX), and *F. angustifolia* x *F. excelsior* (HYB). *F. ornus* (FOR) was used as outgroup. Manna ash genetic profiles collected in the South of Italy are highlighted (in pink) in the phylogenetic tree.

**Figure 3 plants-09-01035-f003:**
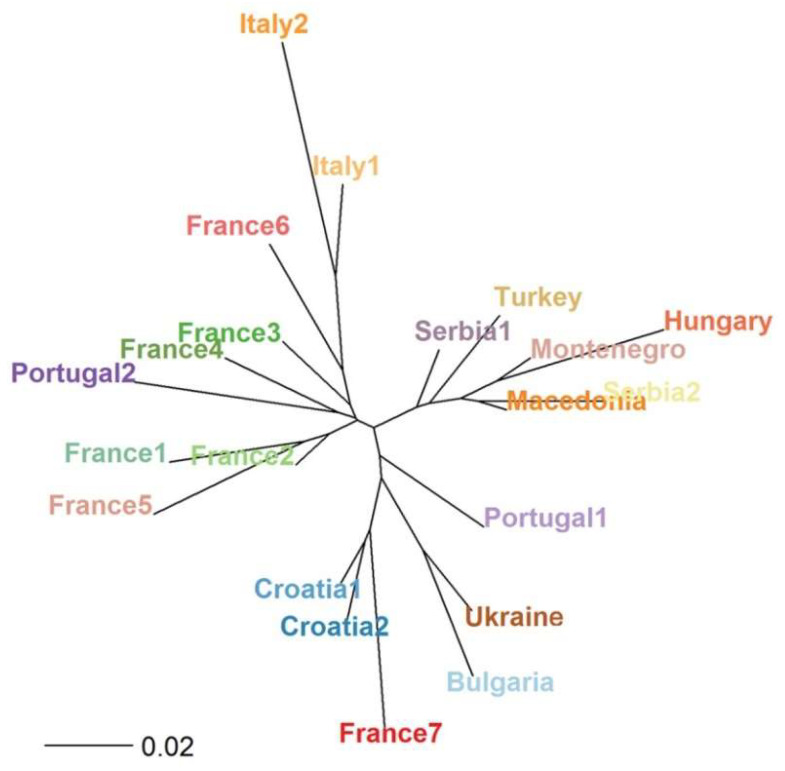
Neighbor-joining (NJ) tree based on population pairwise fixation index. Genetic distances were computed among populations grouped based on sampling locations.

**Figure 4 plants-09-01035-f004:**
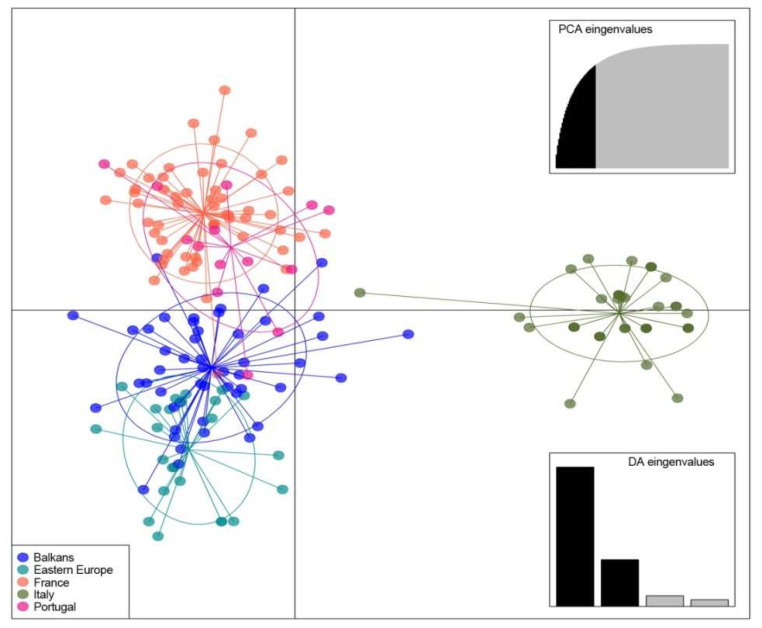
DAPC clustering of European germplasm of *F. angustifolia* studied, using the first two principal components (*Y*-axis and *X*-axis, respectively). The samples are grouped in five main groups: Balkans (blue; Croatia, Macedonia, Montenegro, Serbia, and Turkey), Eastern Europe (cyan; Bulgaria, Hungary, and Ukraine), France (orange), Italy (green) and Portugal (pink).

**Table 1 plants-09-01035-t001:** Genome sizes and morphological traits of studied samples, belonging to manna ash collection. *F. ornus* was used as reference. In bracket the number of samples analyzed belonging to each local variety/species. Three replicates for each trait and the genome sizes evaluation were analyzed.

Type	Local Varieties	Ploidy Level	2C DNA Value *	Apical Leaf (Length/Width Ratio) *	Samara Stalk Length (cm) *	Number of Samaras/Raceme *	Flowering Time	Apical Buds Colour
Manna ash	Abbassa cappeddu (3)	2n	1.540 ± 0.01	0.21 ± 0.01	0.81 ± 0.07	5.51 ± 0.20	November–January	Red
	Baciciu (4)	2n	1.544 ± 0.03	0.42 ± 0.02	1.12 ± 0.07	3.50 ± 1.02	November–January	Red
	Cavolo (3)	2n	1.542 ± 0.02	0.23 ± 0.09	2.11 ± 0.06	5.50 ± 0.07	November–January	Red
	Macigna (3)	2n	1.541 ± 0.01	0.42 ± 0.03	0.79 ± 0.02	6.01 ± 0.70	November–January	Red
	Nivuru (3)	2n	1.541 ± 0.04	0.35 ± 0.01	3.04 ± 0.01	7.10 ± 0.62	November–January	Red
	Nsiriddu (3)	2n	1.544 ± 0.03	0.24 ± 0.08	1.15 ± 0.05	5.09 ± 0.07	November–January	Red
	Russu (3)	2n	1.542 ± 0.04	0.32 ± 0.07	0.68 ± 0.07	5.52 ± 0.27	November–January	Red
	Sarvaggio (3)	2n	1.539 ± 0.02	0.31 ± 0.05	0.77 ± 0.02	5.01 ± 0.19	November–January	Red
	Verdello (7)	2n	1.542 ± 0.05	0.21 ± 0.01	0.76 ± 0.08	6.53 ± 0.35	November–January	Red
	Frassino monumentale (1)	2n	1.541 ± 0.01	0.23 ± 0.03	0.92 ± 0.03	7.05 ± 0.10	November–January	Red
	Nsiriddu monumentale (1)	2n	1.544 ± 0.01	0.22 ± 0.01	0.91 ± 0.05	7.03 ± 0.32	November–January	Red
	*mean*	*-*	*1.542 ***	*0.28 ***	*1.18 ***	*5.80 ***	*-*	*-*
*F. ornus* (4)		2n	1.98 ± 0.01	0.52 ± 0.02	0.41 ± 0.03	46.02 ± 2.05	May–June	White

* Mean values. ** Significantly different at the 0.01 probability level between local varieties and *F. ornus*.

**Table 2 plants-09-01035-t002:** Haplotypes detected based on fragment length polymorphism using six chloroplast microsatellites (cpSSR) and three plastid DNA regions in the germplasm belonging to the studied manna ash collection. *F. ornus* was used as reference. In bracket the number of samples analyzed belonging to each local variety/species.

Samples/Local Varieties	N_h_ ^#^	ccmp2	ccmp3	ccmp4	ccmp6	ccmp7	ccmp10	CPFRAX6	atpB/rbcL	matK	Haplotypes *
Abbassa cappeddu (3)	1	194	97	140	98	117	104	365	157	253	H10
Baciciu (4)	1	194	97	140	98	117	103	365	157	253	H5
Cavolo (3)	1	194	97	140	98	117	103	365	157	253	H5
Macigna (3)	1	194	97	140	98	117	103	365	157	253	H5
Nivuru (3)	1	194	97	140	98	117	103	365	157	253	H5
Nsiriddu (3)	1	194	97	140	98	117	103	366	157	253	H5 ^a^
Russu (3)	1	194	97	140	98	117	103	365	157	253	H5
Sarvaggio (3)	1	194	97	140	98	117	103	365	157	253	H5
Verdello (7)		194	97	140	98	117	103	365	157	253	H5
	3	194	97	140	98	117	103	366	157	253	H5 ^a^
		194	97	140	98	117	104	365	157	253	H10
Frassino monumentale (1)	1	194	97	140	98	117	103	365	157	253	H5
Nsiriddu monumentale (1)	1	194	97	140	98	117	103	365	157	253	H5
*F. ornus* (4)	1	190	97	140	99	117	106	364	157	253	H19

**^#^** Nh, number of haplotypes. * Haplotype classification following Heuertz et al. (2006) [4], without CPFRAX6, atpB/rbcL and matK profiles.^a^ H5 sub-variation identified using also CPFRAX6, atpB/rbcL and matK profiles.

**Table 3 plants-09-01035-t003:** *Fst* values (below diagonal) and *Nei* (1973) genetic distances (above diagonal) evaluated through nSSR on sample set #2. Each parameter was calculated for plants belonging to *F. angustifolia* collected in Italy—FAN (Italy), *F. angustifolia* sampled across Europe—FAN, *F. excelsior*—FEX, and hybrids—HYB (*F. angustifolia* × *F. excelsior*).

Group.	FAN (Italy)	FAN	HYB	FEX
FAN (Italy)	0.000	0.284	0.536	0.874
FAN	0.066	0.000	0.159	0.437
HYB	0.107	0.030	0.000	0.110
FEX	0.163	0.084	0.028	0.000

**Table 4 plants-09-01035-t004:** Summary of genetic variation statistics at 11 nSSR loci on FAN samples belonging to set #2. Individuals were grouped based on sampling location (see Appendix A).

Group ID.	*n*.	*I*	*Ho*	*He*	*Fis*	*AR*
**Bulgaria**	8	0.964	0.615	0.525	−0.208	2.73
**Croatia1**	8	1.156	0.591	0.563	−0.025	3.05
**Croatia2**	8	1.000	0.614	0.523	−0.135	2.71
**France1**	7	1.259	0.645	0.632	−0.030	3.32
**France2**	7	1.320	0.803	0.680	−0.216 *	3.42
**France3**	8	1.316	0.614	0.658	0.039	3.32
**France4**	8	1.171	0.659	0.607	−0.074	3.03
**France5**	8	1.094	0.489	0.572	0.204 **	2.93
**France6**	8	1.256	0.648	0.626	−0.042	3.22
**France7**	8	0.953	0.659	0.513	−0.269	2.63
**Hungary**	7	0.801	0.805	0.506	−0.607 ***	2.28
**Italy1**	16	1.170	0.591	0.553	0.040 ***	2.96
**Italy2**	8	1.032	0.623	0.541	−0.112 ***	2.85
**Macedonia**	8	1.325	0.750	0.651	−0.152	3.39
**Montenegro**	8	1.215	0.659	0.607	−0.087	3.13
**Portugal1**	6	1.185	0.621	0.624	0.017	3.17
**Portugal2**	8	1.029	0.534	0.529	0.038	2.86
**Serbia1**	8	1.323	0.623	0.623	−0.022	3.40
**Serbia2**	6	1.193	0.667	0.580	−0.159	3.26
**Turkey**	3	1.089	0.727	0.601	−0.241	3.55
**Ukraine**	8	1.040	0.610	0.553	−0.109	2.88

*n*. = number of samples for each group; *I* = Shannon’s index; *Ho* = observed heterozygosity; *He* = expected; *Fis* = inbreeding coefficient; *AR =* allelic richness; * *p* < 0.05; ** *p* < 0.01; *** *p* < 0.001.

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
