# Peer review of "Genetic Distinctiveness Highlights the Conservation Value of a Sicilian Manna Ash Germplasm Collection Assigned to Fraxinus angustifolia (Oleaceae)"

_plants, 2020, doi:10.3390/plants9081035_

Round 1
Reviewer 1 Report
This is a very well-conceived and executed study, worthy of publication in Plants. This paper was my introduction to Fraxinus and I appreciated the orientation provided in the introduction. The analytical methods were clearly described and other reviewers will need verify their relevance and application.
The importance of the Sicilian plants was well documented and the expansion of the study to other geographic origins of Fraxinus was appropriate and critical to the conclusions reached.
While the English language usage was very good, some careful editing is required. The readability of the introduction may be improved by separating long paragraphs into smaller, topic-based, paragraphs.
Small corrections are needed in the syntax that may be corrected by the journal's copy editors.
I have a concern about 'collections' and about 'conservation status'. There is minor confusion about collections as being an assemblage of accessions in a genebank of seeds or plants in a permanent garden, or collections of tissues from in situ populations. Here samples were taken in situ from field plantations or natural stands. I suggest that mention be made about permanent identifications by tagging or GPS coordinates of the plants sampled for the diversity/identity studies. Since these plants may be vulnerable to loss for various reasons, I urge that re-propagation to a secure site, such as a botanic garden, be done. Also, have the genomic resources developed in this study been curated in a permanent and secure database or facility?
Finally, I would definitely like to see some discussion about how these valuable plant genetic resources will be permanently conserved in situ and ex situ. How would that be connected to national (Italian) and international conservation programs?
;
Author Response
Dear Editor and Reviewers,
Thank you for your letter and for the reviewers’ comments concerning our manuscript entitled "Genetic distinctiveness highlights the conservation value of a Sicilian manna ash germplasm collection assigned to Fraxinus angustifolia (Oleaceae)”. Those comments are all valuable and very helpful for revising and improving our paper, as well as the important guiding significance to our researches. We have studied comments carefully and have made correction which we hope meet with approval. We have revised the manuscript, according to the comments and suggestions of the referees, and responded, point by point to comments as listed below. All the revised parts have been marked with red font in the revised manuscript and attached files.
We hope that the revised version of the manuscript is now acceptable for publication in your journal.
With kindest regards,
Yours sincerely
Francesco Mercati
Replies to Editor and Reviewers
First of all, we thank both reviewers for positive and constructive comments and suggestions. The following is a point-to-point response to the two reviewers’ comments, as listed below (the answers are in bold and red).
Replies to Reviewer #1
This is a very well-conceived and executed study, worthy of publication in Plants. This paper was my introduction to Fraxinus and I appreciated the orientation provided in the introduction. The analytical methods were clearly described and other reviewers will need verify their relevance and application.
Answer: we would like to thank you in advance for your professional suggestions and comments on our manuscript, and we are glad you liked our manuscript.
The importance of the Sicilian plants was well documented and the expansion of the study to other geographic origins of Fraxinus was appropriate and critical to the conclusions reached.
While the English language usage was very good, some careful editing is required. The readability of the introduction may be improved by separating long paragraphs into smaller, topic-based, paragraphs.
Answer: thank you for your advice; due to the high number of papers available on the topic discussed, before submission we done strong efforts to summarize and clarify all the information showed. Therefore, in our opinion, breaking up the introduction into several paragraphs could distort it. However, we have kept it in a single chapter but, to make it more streamlined and fluid, we have removed some "accessory" parts.
Small corrections are needed in the syntax that may be corrected by the journal's copy editors.
I have a concern about 'collections' and about 'conservation status'. There is minor confusion about collections as being an assemblage of accessions in a genebank of seeds or plants in a permanent garden, or collections of tissues from in situ populations. Here samples were taken in situ from field plantations or natural stands. I suggest that mention be made about permanent identifications by tagging or GPS coordinates of the plants sampled for the diversity/identity studies. Since these plants may be vulnerable to loss for various reasons, I urge that re-propagation to a secure site, such as a botanic garden, be done. Also, have the genomic resources developed in this study been curated in a permanent and secure database or facility?
Answer: thank you for your advice. The material has been taken in situ from natural stands and all plants were tagged. In addition, due to the importance of plant material for conservation and to plan possible future breeding programs, all local varieties have been planted in two repositories in the Madonie Mountains: at the 1) Professional Institute for Agriculture and the Environment (I.p.a.a.) "Luigi Failla Tedaldi", Castelbuono (Italy); and at 2) the private repository of Prof. Schicchi located in the Croce-Foresta district at Castelbuono (Italy). Both repositories are checked and supervised by Prof. Schicchi (email: rosario.schicchi@unipa.it), co-author of the present manuscript.
Finally, I would definitely like to see some discussion about how these valuable plant genetic resources will be permanently conserved in situ and ex situ. How would that be connected to national (Italian) and international conservation programs?
Answer: thank you for your comment. We added in the text the information reported in previous comment, in order to easily access and exchange this important material. In addition, in the present paper has been involved as co-author Myriam Heuertz, a researcher at INRAE (France) that has a strong background on ash. This international network is the starting point to participate in future EU calls, to well use the available biodiversity.

Reviewer 2 Report
Dear authors,
I had the pleasure to review the article entitled: Genetic distinctiveness highlights the conservation value of a Sicilian manna ash germplasm collection assigned to Fraxinus angustifolia (Oleaceae).
The article is well written and organized. I would like to suggest some modifications to improve it.
In my point of view, the Introduction is long: the authors could make an effort to reduce it and make it more readable and less dispersive to lead to the results of this interesting manuscript. The authors could probably remove the part about global warming, in my opinion.
Some references are not correctly indicated and the adverb respectively is often misused.
Line 181: is 4ug/uL the final concentration? If yes, the author should find the correct way to indicate it.
Line 191: the acronyms are not completely clear. Please reformulate the meaning of the acronyms
Line 193: one reference is missing (concerning Harbourne et al 2005)
Results: it is not completely clear the experimental design followed by the authors that omitted some samples of F. angustifolia in the set#1. If the authors have these results, they are strongly encouraged to include them in the manuscript because this can complete the comparison make simple the final analysis.
Line 246: the authors state that some morphological traits ‘were highly variable’. This variability is not actually visible. The authors should support the statement with some statistics or change the sentence, since the most high variability has been observed comparing the Sicilian manna ash germplasm collection and the reference F. ornus. The authors could also improve Table S1 by calculating the mean values as showed in Table 1.
Line 271: The level of polymorphism is low: how can it be significant?
Line 278: what kind of sub-variant are the authors talking about? It would be great to know it
Line 288: What about Macigna? These samples are not grouped altogether, I see.
It would be better to cite Table S2 where needed (in the text and in the Fig.1)
Line 307: the number of the reference should be corrected.
Figure 2 is really disputable. No sample names are visible, even at maximum magnification. And of course, I couldn’t check the veracity of the described results. It will be tough to make all names visible because they are uncountable but the authors should find a better way to present these results. Light blue is not enough distinguishable.
Line 335: below and above diagonal could be better the below and above alone. The authors could alternatively say: Pairwise Fst value etc etc
Line 370: Fis= Fixation index?
Discussion: in several paragraphs of the Results section Verdello is always showing a different behavior from the remaining germplasm collection. What do the authors think about that?
Author Response
Dear Editor and Reviewers,
Thank you for your letter and for the reviewers’ comments concerning our manuscript entitled "Genetic distinctiveness highlights the conservation value of a Sicilian manna ash germplasm collection assigned to Fraxinus angustifolia (Oleaceae)”. Those comments are all valuable and very helpful for revising and improving our paper, as well as the important guiding significance to our researches. We have studied comments carefully and have made correction which we hope meet with approval. We have revised the manuscript, according to the comments and suggestions of the referees, and responded, point by point to comments as listed below. All the revised parts have been marked with red font in the revised manuscript and attached files.
We hope that the revised version of the manuscript is now acceptable for publication in your journal.
With kindest regards,
Yours sincerely
Francesco Mercati
Replies to Editor and Reviewers
First of all, we thank both reviewers for positive and constructive comments and suggestions. The following is a point-to-point response to the two reviewers’ comments, as listed below (the answers are in bold and red).
Replies to Reviewer #2
Dear authors,
I had the pleasure to review the article entitled: Genetic distinctiveness highlights the conservation value of a Sicilian manna ash germplasm collection assigned to Fraxinus angustifolia (Oleaceae).
The article is well written and organized. I would like to suggest some modifications to improve it.
Answer: we would like to thank you in advance for your professional suggestions and comments on our manuscript, and we are glad you liked this manuscript.
In my point of view, the Introduction is long: the authors could make an effort to reduce it and make it more readable and less dispersive to lead to the results of this interesting manuscript. The authors could probably remove the part about global warming, in my opinion.
Answer: thank you for your advice. We modified the introduction following your suggestions.
Some references are not correctly indicated and the adverb respectively is often misused.
Answer: thank you for your advice. We have checked all reference and corrected all mistakes.
Line 181: is 4ug/uL the final concentration? If yes, the author should find the correct way to indicate it.
Answer: the correct concentration is 4ug/ml, as reported in the text, following the protocol described in Otto, 1995.
Line 191: the acronyms are not completely clear. Please reformulate the meaning of the acronyms
Answer: thank you for your advice. We checked and explained well all acronyms.
Line 193: one reference is missing (concerning Harbourne et al 2005)
Answer: thank you for your advice. We corrected the mistake.
Results: it is not completely clear the experimental design followed by the authors that omitted some samples of F. angustifolia in the set#1. If the authors have these results, they are strongly encouraged to include them in the manuscript because this can complete the comparison make simple the final analysis.
Answer: thank you for your advice. Probably this question is related to the set#1 merged with the wide Fraxinus spp collection, published in Gerard et al. (2013) [44]. As reported in the text, to avoid redundancy in the results, we used in set#2 only the samples showing unique SSR profile. Therefore, on the basis of Figure 1 and Table S2, since from 34 samples belonging to manna ash collection we isolated 16 different SSR profiles, we included in subsequent analyzes 16 samples/profiles. However, we modified the sentence to make more fluid and understandable the text.
Line 246: the authors state that some morphological traits ‘were highly variable’. This variability is not actually visible. The authors should support the statement with some statistics or change the sentence, since the most high variability has been observed comparing the Sicilian manna ash germplasm collection and the reference F. ornus. The authors could also improve Table S1 by calculating the mean values as showed in Table 1.
Answer: thank you for your advice. The text has been modified following your suggestions. In addition Table S1 has been improved.
Line 271: The level of polymorphism is low: how can it be significant?
Answer: About the plastidial markers, the low polymorphic rate was expected due to known low variability of chloroplast sequences (as reported in many papers published).
Line 278: what kind of sub-variant are the authors talking about? It would be great to know it
Answer: the sub-variant was already highlighted in Table 2. We compared the cpSSR profiles recorded across the ash manna collection to the available profiles reported in Heuertz et al. (2006) [4]. Using the ccmp set, we isolated two known profiles (H5 and H10) already described in F. angustifolia. In addition, to characterize our collection we used also additional primers (CPFRAX6, atpB/rbcL and matK) and, among these, the values recorded for CPFRAX6 underlined the sub-variant targeted as H5a. Unfortunately, we cannot compare these last markers to other available profiles therefore, we decide to indicate the different profile as sub-variants of H5 haplotype. However, there was some errors in the notes, we checked and modified all mistakes.
Line 288: What about Macigna? These samples are not grouped altogether, I see.
Answer: thank you for your advice. Yes, the comment is correct; we modified the sentence, adding also Macigna.
It would be better to cite Table S2 where needed (in the text and in the Fig.1)
Answer: thank you for your advice. We cited Table S2 where needed.
Line 307: the number of the reference should be corrected.
Answer: thank you for your advice. As previous reported, we corrected all mistakes.
Figure 2 is really disputable. No sample names are visible, even at maximum magnification. And of course, I couldn’t check the veracity of the described results. It will be tough to make all names visible because they are uncountable but the authors should find a better way to present these results. Light blue is not enough distinguishable.
Answer: the aim of this figure was to underline the strong differences among Fraxinus species (marked with different colors), and to highlighting the distinctive clustering of Southern Italian F. angustifolia germplasm; therefore it was important a clear representation for each group/species studied. Due to the high number of samples analyzed it was impossible show clearly the names of each sample. In addition, the data reported in figure 2 were also supported by Table 3, Table 4, Table S2, Table S4 and Figure 3. However, we improved (as possible) the resolution of Figure 2, highlighting better the samples collected in Southern Italy. Now, in the version inserted increasing the zoom the names are readable.
Line 335: below and above diagonal could be better the below and above alone. The authors could alternatively say: Pairwise Fst value etc etc
Answer: we modified the caption following your suggestion.
Line 370: Fis= Fixation index?
Answer: thank you for your advice. Fis is the Inbreeding coefficient, we modified this note.
Discussion: in several paragraphs of the Results section Verdello is always showing a different behavior from the remaining germplasm collection. What do the authors think about that?
Answer: thank you for your observation. As reported in the text, due to the high number of samples, in comparison to the other local varieties, there is higher genetic variability in this variety than others. This is an interesting point for future breeding programs finalized to select more productive genotypes among Verdello that is actually the most represented local variety and used for manna production.
